# Magnetic Resonance Relaxometry for Tumor Cell Density Imaging for Glioma: An Exploratory Study via ^11^C-Methionine PET and Its Validation via Stereotactic Tissue Sampling

**DOI:** 10.3390/cancers13164067

**Published:** 2021-08-12

**Authors:** Manabu Kinoshita, Masato Uchikoshi, Souichiro Tateishi, Shohei Miyazaki, Mio Sakai, Tomohiko Ozaki, Katsunori Asai, Yuya Fujita, Takahiro Matsuhashi, Yonehiro Kanemura, Eku Shimosegawa, Jun Hatazawa, Shin-ichi Nakatsuka, Haruhiko Kishima, Katsuyuki Nakanishi

**Affiliations:** 1Department of Neurosurgery, Osaka International Cancer Institute, 3-1-69 Otemae, Chuo-ku 541-8567, Japan or masato1.uchikoshi@medical.canon (M.U.); tomohikoozaki@gmail.com (T.O.); mm0001ak@yahoo.co.jp (K.A.); fujiita220@gmail.com (Y.F.); takahiro.matsuhashi@gmail.com (T.M.); 2Canon Medical Systems Corporation, 1385 Shimoishigami, Otawara-shi 324-8550, Japan; 3Department of Diagnostic and Interventional Radiology, Osaka International Cancer Institute, 3-1-69 Otemae, Chuo-ku 541-8567, Japan or souichirou.tateishi@oici.jp (S.T.); or shohei.miyazaki@oici.jp (S.M.); s-mio@umin.ac.jp (M.S.); je2k-nkns@asahi-net.or.jp (K.N.); 4Department of Biomedical Research and Innovation, Institute for Clinical Research, National Hospital Organization Osaka National Hospital, 2-1-14 Hoenzaka, Chuo-ku 540-0006, Japan; yonehirok@gmail.com or; 5Department of Nuclear Medicine and Tracer Kinetics, Osaka University Graduate School of Medicine, 2-2 Yamadaoka, Suita 565-0871, Japan; eku@tracer.med.osaka-u.ac.jp (E.S.); hatazawa@tracer.med.osaka-u.ac.jp (J.H.); 6Department of Diagnostic Pathology and Cytology, Osaka International Cancer Institute, 3-1-69 Otemae, Chuo-ku 541-8567, Japan; ZAT02544@nifty.ne.jp; 7Department of Neurosurgery, Osaka University Graduate School of Medicine, 2-2 Yamadaoka, Suita 565-0871, Japan; hkishima@nsurg.med.osaka-u.ac.jp

**Keywords:** malignant glioma, ^11^C-methionine positron emission tomography (MET-PET), magnetic resonance imaging (MRI), relaxometry, non-contrast-enhancing tumor

## Abstract

**Simple Summary:**

To test the hypothesis that quantitative magnetic resonance relaxometry reflects glioma tumor load within tissue and that it can be an imaging surrogate for visualizing non-contrast-enhancing tumors, we investigated the correlation between T1- and T2-weighted relaxation times, apparent diffusion coefficient (ADC) on magnetic resonance imaging, and ^11^C-methionine (MET) on positron emission tomography (PET). Moreover, we compared T1- and T2-relaxation times and ADC with tumor cell density (TCD) findings obtained via stereotactic image-guided tissue sampling. A T1-relaxation time of >1850 ms but <3200 ms or a T2-relaxation time of >115 ms but <225 ms under 3 T indicated high MET uptake. The stereotactic tissue sampling findings confirmed that the T1-relaxation time of 1850–3200 ms significantly indicated higher TCD while the T2-relaxation time and ADC did not significantly correlate with the stereotactic tissue sampling findings. However, synthetically synthesized tumor load images from the T1- and T2-relaxation maps were able to visualize MET uptake presented on PET.

**Abstract:**

One of the most crucial yet challenging issues for glioma patient care is visualizing non-contrast-enhancing tumor regions. In this study, to test the hypothesis that quantitative magnetic resonance relaxometry reflects glioma tumor load within tissue and that it can be an imaging surrogate for visualizing non-contrast-enhancing tumors, we investigated the correlation between T1- and T2-weighted relaxation times, apparent diffusion coefficient (ADC) on magnetic resonance imaging, and ^11^C-methionine (MET) on positron emission tomography (PET). Moreover, we compared the T1- and T2-relaxation times and ADC with tumor cell density (TCD) findings obtained via stereotactic image-guided tissue sampling. Regions that presented a T1-relaxation time of >1850 ms but <3200 ms or a T2-relaxation time of >115 ms but <225 ms under 3 T indicated a high MET uptake. In addition, the stereotactic tissue sampling findings confirmed that the T1-relaxation time of 1850–3200 ms significantly indicated a higher TCD (*p* = 0.04). However, ADC was unable to show a significant correlation with MET uptake or with TCD. Finally, synthetically synthesized tumor load images from the T1- and T2-relaxation maps were able to visualize MET uptake presented on PET.

## 1. Introduction

Both low- and high-grade gliomas are known to invade into surrounding normal brain parenchyma. Main tumor masses have classically been recognized as regions showing contrast enhancement on magnetic resonance imaging (MRI) while surrounding non-enhancing lesions with an abnormal appearance were considered to represent infiltration of the tumor into the brain parenchyma. The so-called “gross total removal” of the tumor is usually defined for situations in which surgical removal of the entire contrast-enhancing lesion is achieved. Furthermore, conventional radiation therapy is planned based on the assumption that contrast-enhancing regions represent the tumor core. At the same time, non-contrast-enhancing lesions are merely regions with tumor cell infiltration, allowing radiation dose reductions in those areas. In recent years, however, non-contrast-enhancing lesions were shown to potentially harbor extremely heavy tumor load, with cell densities reaching almost as high as that in the enhancing tumor core [1,2,3]. This problem was proven by stereotactic intra-operative tissue sampling, suggesting that tumor cell density (TCD) within the non-contrast-enhancing regions should not be ignored [3,4]. Although numerous attempts have been made to visualize the magnitude of tumor cell load non-invasively [5,6], amino acid positron emission tomography (PET) currently seems to be the only reliable imaging modality to fulfill this requirement. Regardless of the imaging sequence chosen, MRI is still considered inferior to PET in this regard [3,5,6,7,8,9,10]. In this current study, we tested the hypothesis that quantitative T1- or T2-weighted relaxometry reflects the extent of glioma tumor load, which could then be used to estimate and visualize the non-contrast-enhancing areas of heavy tumor load. This hypothesis was tested in two stages: first, by investigating the relationship between T1- or T2-relaxation times and ^11^C-methionine (MET) uptake on PET, and second, by comparing the T1- or T2-relaxation times with the TCD findings obtained via stereotactic image-guided tissue sampling. Furthermore, we also evaluated the capability of apparent diffusion coefficient (ADC) derived from diffusion-weighted images (DWI) to predict TCD, as this metric is often discussed as a promising imaging surrogate of TCD in glioma [11].

## 2. Materials and Methods

### 2.1. Patient Selection

The local institutional ethical review board approved the use of clinical data for this research, and each patient provided written informed consent (approval number 1612065191). Low- or high-grade glioma patients, who underwent both T1- and T2-relaxometry as pre-surgical imaging studies, were included for analysis. In total, 22 patients met this criterion. We performed MET-PET in 10 patients and diffusion-weighted imaging (DWI) in 17 patients. Stereotactic-image-guided tissue sampling was carried out intraoperatively in all but 1 patient, and 79 samples were obtained. Pathological diagnosis was based on the 2016 World Health Organization classification for central nervous system tumors (Figure 1 and Table 1).

### 2.2. T1- and T2-Relaxometry

Imaging was performed on either a 1.5 or 3 T MR scanner (Prisma or Aera; Siemens Healthcare, Erlangen, Germany). T1-relaxometry was achieved by first acquiring MP2RAGE images, then converting those images into T1-relaxation time maps. At the same time, T2-relaxometry was achieved by first acquiring multi-echo T2-weighted images and then converting those images into T2-relaxation time maps, with both relaxometries performed via Bayesian inference modeling (Olea Nova+; Canon Medical Systems, Tochigi, Japan) [12]. T1- and T2-relaxation times measured on 1.5 T MRI were then converted into estimated relaxation times measured at 3 T by a conversion coefficient using data from a normal volunteer as follows (Appendix A).
T_1@3T_ = 1.19 × T_1@1.5T_(1)
T_2@3T_ = 0.92 × T_2@1.5T_(2)

All relaxation time data presented hereafter are provided as measurements performed under 3-T. MP2RAGE was acquired using: repetition time (TR) = 5000 ms; echo time (TE) = 3.86 ms; and inversion time (TI) = 935/2820 ms. Multi-echo T2-weighted imaging was acquired using TR = 4000 ms; TE = 20, 40, 60, 80, 100, 120, and 140 ms for both 1.5-T and 3-T. An additional 20 min scan was necessary on top of routine clinical imaging of glioma patients.

### 2.3. ^11^C-Methionine Positron Emission Tomography (MET-PET)

PET studies were performed using an Eminence-G system (Shimadzu, Kyoto, Japan), with MET synthesized according to the method described by Hatakeyama et al. [13] and injected intravenously at a dose of 3 MBq/kg body weight. Tracer accumulation was recorded for 12 min in 59 or 99 transaxial sections over the entire brain. Summed activity from 20 to 32 min after tracer injection was used for image reconstruction. Images were stored in 256 × 256 × 59 or 99 anisotropic voxels, with each voxel being 1 × 1 × 2.6 mm. An area of high cell density was defined as those voxels presenting a tumor-to-normal tissue ratio (T/N ratio) >1.5. This cut-off was derived from previous publications showing that the T/N ratio = 1.5 was roughly equivalent to tissues with a cell density of 2000 cells/mm^2^ [3,6,14,15]. As cell density of healthy brain tissue ranges from 382 to 1106 cells/mm^2^, this cut-off was considered the most appropriate for defining with confidence in those regions carrying a high tumor load [15].

### 2.4. Diffusion-Weighted Imaging (DWI) and Apparent Diffusion Coefficient (ADC) Measurement

DWI was performed as part of neurite orientation dispersion and density imaging (NODDI). Images were acquired using a single-shot echo-planar imaging technique, with TE = 80 ms and TR = 5000 ms for 3 T, and TE = 102 ms and TR = 6000 ms for 1.5 T. Diffusion gradient was encoded in 64 directions with *b* = 1000 and 2000 s/mm^2^ along with an additional measurement without the diffusion gradient (*b* = 0 s/mm^2^). A parallel imaging technique was used to record data with a spatial resolution of 128 × 128 and a field of view of 230 × 230 mm. A total of 44 sections were obtained, with a slice thickness of 3.5 mm and no intersection gap. ADC maps were processed using the Diffusion Toolkit (Martinos Center for Biomedical Imaging, Massachusetts General Hospital, Boston, MA, USA; Available online: http://www.trackvis.org/dtk/ (accessed on 29 July 2019) only using DWI obtained with *b* = 1000 s/mm^2^.

### 2.5. Image Fusion/Registration and Stereotactic Image-Guided Tissue Sampling

MET-PET, ADC maps, and T1-, T2-relaxation time maps were co-registered and resliced to a resolution of 256 × 256 × 48, with each voxel size being 0.9 × 0.9 × 3.3 mm. Images were registered using Vinci image-analyzing software (Max-Planck Institute for Neurological Research Cologne, Cologne, Germany; Available online: http://www.nf.mpg.de/vinci/ (accessed on 29 July 2019), allowing voxel-by-voxel analysis of different imaging modalities (Figure 1) [16]. Voxels-of-interest were created based on T2-Fluid-attenuated inversion recovery (FLAIR) images under the assumption that hyperintense regions were suggestive of tumor involvement in tissues. For cases where stereotactically sampled tissues were available, T1- and T2-relaxation time maps were co-registered to the preoperative thin-slice, contrast-enhanced, T1-weighted images using the same workflow described above except for resolution reslicing. We performed stereotactic tissue sampling under craniotomy with a neuronavigation system (BrainLab CURVE; BrainLab, Munich, Germany) assisting the surgery. Surgeries were performed either under general anesthesia or awake craniotomy [17]. We performed biopsies at the earliest stage of operation to minimize the effects of brain-shift during resection. Visual confirmation of anatomical landmarks, such as the cortical veins and sulci, verified the accuracy of the navigation system. A total of 79 tissues were randomly sampled in this manner. Tissue sampling was performed to obtain both contrast-enhancing and non-contrast-enhancing tumor tissues. Finally, 25 samples were obtained from contrast-enhanced lesions, and 54 samples were from non-contrast-enhancing lesions. The sampling locations were stored within the navigation system upon stereotactic tissue sampling as Digital Imaging and Communications in Medicine (DICOM) coordinates of the thin-slice, contrast-enhanced, T1-weighted images. These DICOM coordinates were then converted into the corresponding DICOM coordinates of the T1- and T2-relaxation time maps. Average T1- and T2-relaxation times within a 1-cm^3^ VOI at the target were obtained using software developed in-house and written in MATLAB R2016b (MathWorks, Natick, MA, USA; Appendix A).

### 2.6. Histopathological Analysis

Formalin-fixed specimens were embedded in paraffin for histopathological examination. Hematoxylin and eosin-stained specimens were evaluated to calculate the TCD. Cell counting was performed at ×400 magnification under light microscopy (Nikon, Tokyo, Japan). All cells were counted, except those that were different from tumor cells, such as endothelial cells or lymphocytes. The area for the tumor cell count was 0.0497 mm^2^, and the mean TCD was recorded from 3 different locations within each specimen.

### 2.7. Statistical Analysis

Statistical analysis was performed for two-group comparisons using GraphPad Prism 5 software (GraphPad Software, San Diego, CA, USA). Unpaired t-testing under an assumption of equal variance was used with a threshold of *p* < 0.05 considered as statistically significant. Histogram analysis was also performed using Prism 5 software. A probability histogram was plotted for T1- and T2-relaxation time and ADC. Two histograms were plotted: one for high MET uptake regions (Met-PET high) and another for low MET uptake (Met-PET low). Subsequently, we defined the likeliness of high MET uptake at a given bin *k* as follows;
(3)$$\mathrm{Likeliness}\mathrm{of}\mathrm{Met}-\mathrm{PET}\mathrm{high}(k)=\left\{\frac{n_{H}\left(k\right)}{N_{H}}-\frac{n_{L}\left(k\right)}{N_{L}}\right\}/\left\{\frac{n_{H}\left(k\right)}{N_{H}}+\frac{n_{L}\left(k\right)}{N_{L}}\right\}$$
where *k* denotes a given bin of the histogram, $n_{H}\left(k\right)$ is the number of Met-PET high voxels for bin *k*, $n_{L}\left(k\right)$ is the Met-PET low voxel count for bin *k*, $N_{H}$ is the total number of high MET uptake voxels, and $N_{L}$ is the total number of low MET uptake voxels. A total of 271,650 voxels from 10 patients were analyzed. $N_{H}$ was 99,773 and $N_{L}$ was 171,877 throughout the analysis. This approach is similar to that proposed for analyzing flow cytometric data when comparing two histograms to discriminate data points of interest from negative controls [18,19]. Our method is more similar to the method described by Overton [18] than by Agnès et al. [19], as subtraction of frequency rather than cumulatively counted histogram was performed. However, our method is different from Overton’s as our analysis did not ignore negative values and the obtained value “likeliness of Met-PET high (*k*)” was adjusted for each bin *k*. We took this approach, as firstly, mere subtraction of the number of data points within a bin *k* will not account for the difference in the total sizes of Met-PET high and low voxels (namely $N_{H}$ and $N_{L}$). Secondly, we intended to estimate the likeliness (or likelihood) of a region fitting the requirements for a given bin *k* to be Met-PET high or low. Thus, it was necessary to divide the difference of the two frequencies by the sum of them (Equation (3)). In our case, if the likeliness of Met-PET high (*k*) is a positive value, that will suggest that the bin *k* is more likely to be Met-PET high, while a negative value indicates it to be more likely Met-PET low. The obtained values were plotted as a function of T1-, T2-relaxation times and ADC. The bin width was arbitrarily set as 100 ms for T1- and 5 ms for T2-relaxation time and 0.0005 mm^2^/s for ADC (Appendix A). The bin width was set at the point where the separation of two histograms was visually good. Furthermore, data deriving from bins where the sum of the histograms of Met-PET low and high were lower than 5% of the expected value (that is 0.1 divided by the number of bins) were excluded, as data deriving from these bins were considered unreliable due to insufficient data points.

## 3. Results

### 3.1. Areas of High and Low MET Uptake Show Different T1- and T2-Relaxation Times

Frequency histograms of the voxels with high and low MET uptake on PET were plotted as a function of T1- and T2-relaxation time (Figure 2A,B). Histograms for both T1- and T2-relaxation time for high MET uptake shifted to the right, and widths were narrower than for low-uptake regions (Figure 2A,B). The likeliness of high MET uptake revealed that T1-relaxation time >1900 ms but <3000 ms was indicative of high MET uptake (Figure 2D). Similarly, T2-relaxation time >115 ms but <225 ms was indicative of high MET uptake (Figure 2E). The likeliness of high MET uptake was also noted to differ according to the T1- or T2-relaxation time of interest. These observations suggested that T1- and T2-relaxation time can be used as a surrogate imaging measurement to predict the probability of each voxel of interest showing high or low MET uptake.

### 3.2. ADCs within Areas of High and Low MET Uptake

The performance of ADC to discriminate areas of high and low MET uptake was tested similar to the method described above using the same ten patients. Contrary to our expectation, ADC for high MET uptake shifted to the right (Figure 2C). In theory, high ADC indicates low cell density; therefore, ADC for high MET uptake should have shifted to the left. The likeliness of high MET uptake revealed that ADC > 0.00095 mm^2^/s but <0.00145 mm^2^/s was indicative of high MET uptake (Figure 2F).

### 3.3. Histological Validation of T1- and T2-Relaxation Times and ADC as Surrogate Measurements for TCD

We then attempted to validate the observations mentioned above by stereotactically obtained tissues. The correlations of TCD within the sampled tissues and T1- and T2-relaxation time and ADC are shown in Figure 3. Firstly, Figure 3A confirms that the tissue’s MET uptake and TCD correlate (*p* = 0.02). Next, the correlation of T1-relaxation time and TCD showed similar trends to those already depicted by the MET-PET and T1-relaxation time study (Figure 2A,B). TCDs obtained from locations with T1-relaxation time within the range of 1850–3200 ms were significantly higher than those obtained from other locations (Figure 3B, *p* = 0.04). T2-relaxation time, however, did not exhibit a significant correlation with the TCD of the stereotactically obtained tissues (Figure 3C). ADC showed the worst performance regarding the relationship of TCD with a relatively high *p*-value (Figure 3D).

### 3.4. Synthetic Tumor Load Image via T1- and T2-Relaxometry

As the probability of each voxel being classified as high tumor load can be predicted using T1-relaxation maps using the histogram presented in Figure 2D, a high tumor load probability map can be synthesized (i.e., high MET uptake). Figure 4A shows a representative case of glioblastoma, which exhibits areas of both non-contrast-enhancing tumors and pure vasogenic edema (red asterisk). The T1-relaxation map (Figure 4B) was synthesized into a tumor load probability map (Figure 4C). Comparing with MET-PET (Figure 4D), the synthesized tumor load probability map was able to discriminate two different types of T2-hyperintense lesions, i.e., one due to non-contrast-enhancing tumor and another due to pure vasogenic edema.

## 4. Discussion

One of the most crucial yet challenging issues for glioma patient care is visualizing non-contrast-enhancing tumor regions [4,6,20,21,22,23]. Numerous studies using magnetic resonance spectroscopy, amino-acid PET, or stereotactic tissue sampling have shown that diseased tissues extend far beyond the enhancing regions. T2-hyperintense lesions were conventionally considered as brain tissue “invaded” by tumor cells. It is now, however, considered that these lesions could be categorized into two types of pathology: (1) classical tumor cells invading brain tissue with little tumor load, and (2) non-contrast enhancing but heavily tumor cell-loaded “non-enhancing tumor tissue” [1,3,4]. Errors in discriminating between these two conditions can lead to under-treatment of high-grade glioma patients, which could, in turn, lead to a poor prognosis [4]. Methods of providing suitable imaging contrast to visualize non-enhancing tumor tissues are thus required.

The possibility of using MR relaxometry for glioma cell density prediction was first pursued in an animal study [24], followed by a study in patients by Ellingson et al. [2,25]. T2-hyperintense regions were classified as either non-enhancing tumor (NET) or edema by radiological reading, and cut-offs of 125–250 ms for NET and 250–500 ms for edema as measured by T2-relaxation time were determined. Of note, T2-relaxation time was measured by the dual-echo turbo spin-echo sequence for that study, differing from the multi-echo sequence method used in the present study, and yet the proposed NET volume was prognostic for glioblastoma patients [2]. Intriguingly, this observation is generally in line with that of the current research revealing that a T2-relaxation time of 115–225 ms was indicative of heavily tumor-loaded tissue as evaluated by MET uptake (Figure 2E). These results, in combination, could lead to the conclusion that a T2-relaxation time of approximately 115–225 ms measured under 3 T is the optimal threshold to extract tumor tissues with heavily loaded tissue within the NET. However, T1-relaxation time was a much better image surrogate for TCD within a tissue. The presented data demonstrated a more robust performance for predicting heavily tumor-loaded tissues than T2-relaxation time (Figure 2 and Figure 3). This result might be due to the broader dynamic range of T1-relaxation time compared to that of T2-relaxation time. As the T1-relaxation time for cerebrospinal fluid is 4000 ms, a low probability of heavy tumor load with T1-relaxation time >3200 ms could be attributable to contamination of fluid-rich tumor tissues such as cystic lesions. On the other hand, the low probability of a heavy tumor load with T1-relaxation time <1850 ms that shows high intensity on T2-weighted imaging could be considered as typical of “tumor cell-invaded brain edema with little tumor load” or “vasogenic edema.” Furthermore, the current research was able to propose the use of the T1-relaxation map for visualizing tumor load in gliomas (Figure 4), which could be used alternative to metabolic imaging modalities such as amino-acid PET. A prospective study is warranted to validate our findings.

Finally, DWI was critically evaluated for its capability in estimating tumor load within the brain, as it has long been expected as a valuable imaging tool [3,11,21,26,27,28,29,30,31]. However, the findings in this current study clearly showed that ADC derived from DWI did not hold up a satisfactory performance in this means. This result is similar to our previous publication using a different cohort [3]. Furthermore, although ADC should theoretically show an inverse correlation with the cell density of the tissue, stereotactically biopsied tissues do not support this hypothesis unless the TCD is extremely high such as for malignant lymphoma [32]. This problem is possibly attributed to the contamination of necrotic and cystic tissues within high-grade lesions, obscuring the correlation between ADC and TCD.

Technical limitations should be addressed in the end. The histogram analysis that we performed should be carefully interpreted, as subtracting frequency histograms and further adjusting each bin to estimate a “likeliness” of a phenomenon of interest is not an analytical method fully accepted in the community. In addition, analysis relying on the histogram is affected by the bin width setting. The presented concept requires external validation for further technological development and usage of magnetic resonance relaxometry for tumor cell density imaging in glioma patients.

## 5. Conclusions

T1-relaxation time (but not T2-relaxation time) and ADC correlated well with TCD in glioma tissues. The ideal ranges for identifying tissue showing high tumor load were 1850–3200 ms in T1-relaxation measured at 3 T. Further incorporation of this method should be considered in glioma imaging following thorough critical evaluation.

## Figures and Tables

**Figure 1 cancers-13-04067-f001:**
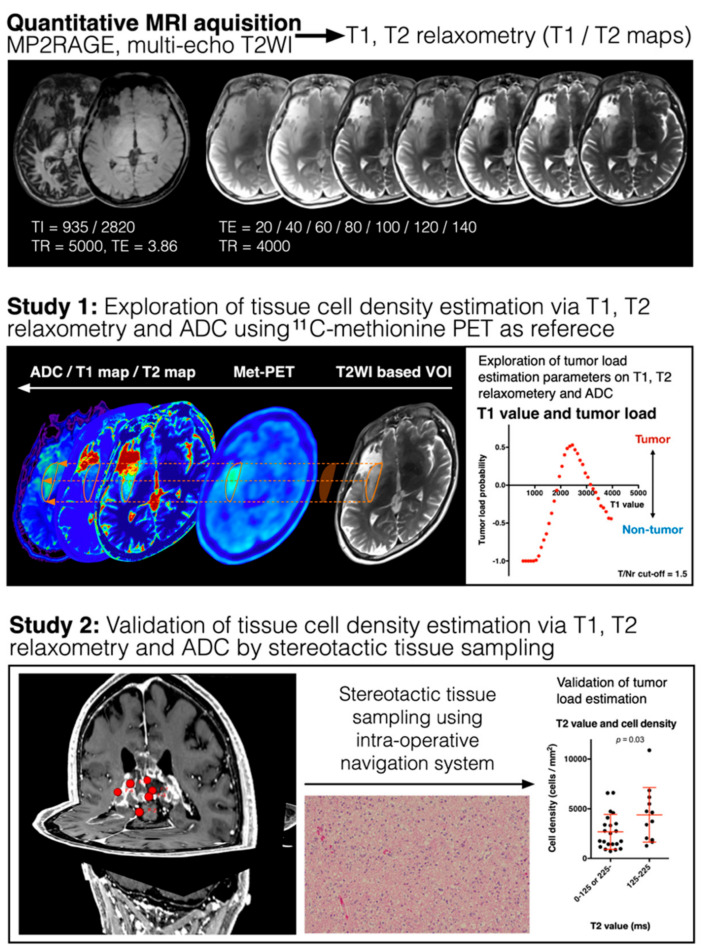
Analytical workflow of the present study. Bayesian inference modeling was utilized to convert MP2RAGE images and multi-echo T2-weighted images to T1-relaxometry and T2-relaxometry. The current study was carried out in two stages: Stage 1, investigating the correlation between T1- or T2-relaxation time and apparent diffusion coefficient (ADC) and ^11^C-methionine uptake as measured by positron emission tomography (Study 1), and Stage 2, comparing T1- or T2-relaxation times and ADC with tumor cell density as measured by stereotactic image-guided tissue sampling (Study 2).

**Figure 2 cancers-13-04067-f002:**
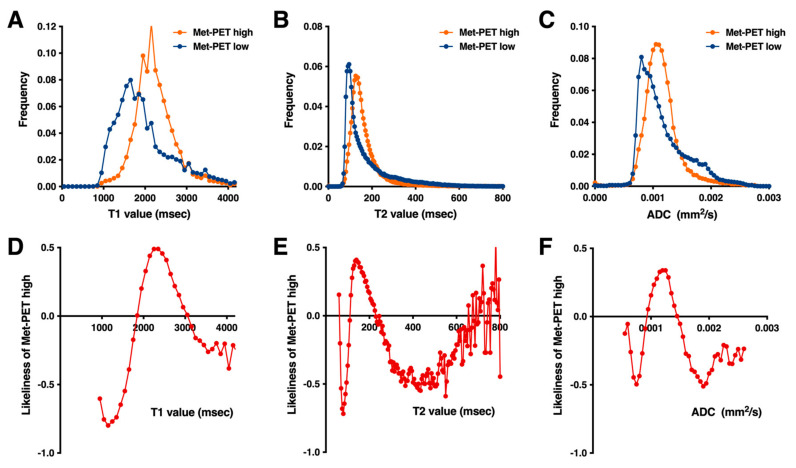
Correlation study between T1-, T2-relaxation times, ADC, and ^11^C-methionine (MET) uptake. Frequency histograms of high MET uptake (T/N ratio >1.5; MET-PET high) and low (T/N ratio ≤1.5; MET-PET low) are presented as a function of T1- (**A**), T2-relaxation time (**B**), and ADC (**C**). Note that MET-PET high uptake skewed to the right, and widths were narrower than MET-PET low uptake in both T1- and T2-relaxation time. The likeliness of high MET uptake, as defined in the text, for T1- (**D**), T2-relaxation time (**E**), and ADC (**F**) are presented. These three graphs show the likeliness of MET-PET high or low uptake as a function of T1-, T2-relaxation times and ADC. Source data are provided in Appendix A.

**Figure 3 cancers-13-04067-f003:**
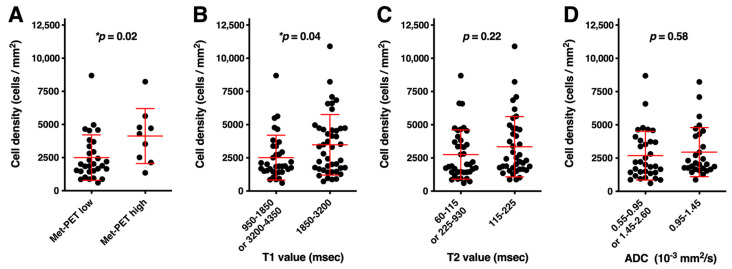
Correlation study between MET uptake, T1-, T2-relaxation times, ADC, and TCD. Cell densities, as measured in stereotactically sampled tissues, are plotted as a function of MET uptake (**A**), T1- (**B**), T2-relaxation time (**C**), and ADC (**D**). MET-PET and T1-relaxation time significantly correlated with TCD (**A**,**B**). Asterisk mark (*) indicates *p* < 0.05.

**Figure 4 cancers-13-04067-f004:**
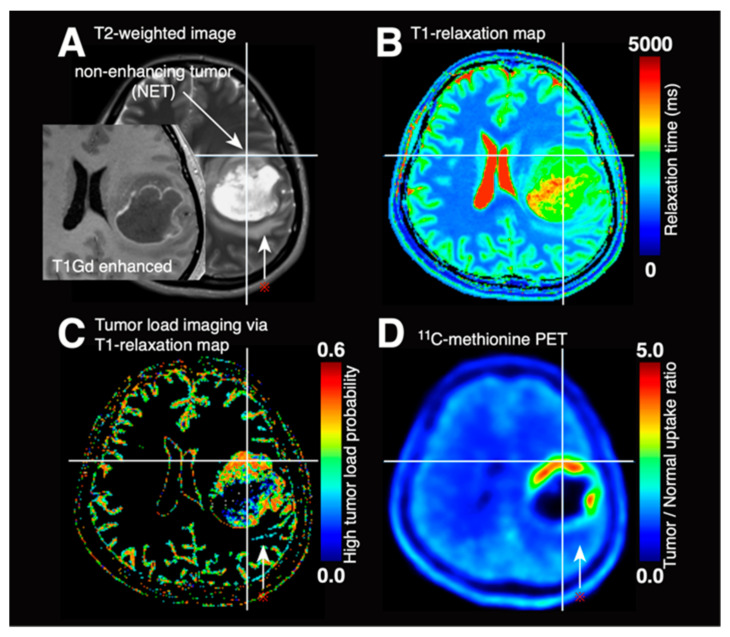
Synthetic tumor load image via T1-relaxometry. A representative case (case #11) of synthetic tumor load imaging is presented. Reference images of contrast-enhanced T1-weighted imaging and T2-weighted imaging are shown (**A**), alongside the raw T1-relaxation map (**B**). High tumor load probability maps are synthesized from T1-relaxation maps (**C**). ^11^C-methionine PET uptake is shown as a reference for ground truth (**D**). It is readily appreciated that there are two types of T2-hyperintense regions and that synthetic tumor load maps correlate well with ^11^C-methionine PET in those areas. The red asterisk indicated T2-hyperintense with a low tumor load.

**Table 1 cancers-13-04067-t001:** Patient characteristics.

#	Sex	Age	Pathology	Molecular Characteristics	MET-PET	ADC	Number of Stereotactic Tissues (Enclosed: Contrast-Enhanced Lesions)		
							T1/T2 map	MET-PET	ADC
1	M	68	AA	IDH-wt	Performed	Performed	4	4	4
2	M	68	AA	IDH-wt	-	Performed	5	-	5
3	F	81	DA	IDH-wt	Performed	Performed	5	5	5
4	M	72	GBM	IDH-wt	-	Performed	3 (3)	-	3 (3)
5	M	19	GBM	IDH-wt	-	-	4 (2)	-	-
6	M	70	GBM	IDH-wt	-	-	6 (3)	-	-
7	F	72	GBM	IDH-wt	-	Performed	1 (1)	-	1 (1)
8	F	68	GBM	IDH-wt	Performed	Performed	2 (2)	2 (2)	2 (2)
9	M	52	GBM	IDH-wt	-	-	3 (2)	-	-
10	M	76	GBM	IDH-wt	-	Performed	3 (2)	-	3 (2)
11	F	34	GBM	IDH-wt	Performed	Performed	7 (6)	7 (6)	7 (6)
12	M	30	rec. AA	IDH-wt	-	-	3	-	-
13	F	45	rec. GBM	IDH-wt	-	Performed	3 (2)	-	3 (2)
14	F	29	OL	IDH-mt, 1p/19q-codeleted	Performed	Performed	6	6	6
15	F	48	OL	IDH-mt, 1p/19q-codeleted	Performed	Performed	4	4	4
16	M	37	OL	IDH-mt, 1p/19q-codeleted	Performed	Performed	-	-	-
17	M	30	OL	IDH-mt, 1p/19q-codeleted	Performed	Performed	4	4	4
18	F	46	rec. AO	IDH-mt, 1p/19q-codeleted	-	-	2 (2)	-	-
19	M	48	rec. AO	IDH-mt, 1p/19q-codeleted	Performed	Performed	3	3	3
20	F	35	AA	IDH-mt	-	Performed	5	-	6
21	M	34	DA	IDH-mt	Performed	Performed	4	4	4
22	F	48	DMG	H3 K27M-mt	-	Performed	1	-	1

rec. = recurrent, AA = Anaplastic astrocytoma, DA = Diffuse astrocytoma, GBM = Glioblastoma, OL = Oligodendroglioma, AO = Anaplastic oligodendroglioma, DMG = Diffuse midline glioma, wt = wildtype, mt = mutant, MET-PET = ^11^C-methionine positron emission tomography, ADC = apparent diffusion coefficient.

## Data Availability

The data presented in this study are available in this article (and Appendix A).

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
