# Peer review of "Magnetic Resonance Relaxometry for Tumor Cell Density Imaging for Glioma: An Exploratory Study via 11C-Methionine PET and Its Validation via Stereotactic Tissue Sampling"

_cancers, 2021, doi:10.3390/cancers13164067_

Round 1
Reviewer 1 Report
The changes introduced in this version of the manuscript, supported by the supplemental materials and the cover letter, seem to make the presentation more clear. Still, there are a few issues which need to be resolved.
First, histogram is a vector. One can subtract vectors, but vectors division (cf. lines 197-198) needs further specification. Perhaps, binwise subtraction and binwise division are meant? If this is the case, the right-side of equation (3) will be
( nH(k)/NH – nL(k)/NL ) / ( nH(k)/NH + nL(k)/NL )
where k is the bin number, nH(k) is the met-PET high voxels count for bin k, nL(k) denotes the met-PET low count for bin k, NH is the total number of met-PET high-uptake voxels, and NL is the total number of met-PET low-uptake voxels.
Second, the values of NH and NL should be explicitly specified. Are they NH=99 773 and NL=171 877 in all three cases?
Third, the left side of equation (3) can not be interpreted as probability. The quantities which are subtracted from each other in the numerator (and added to each other in the denominator) of the right side of this equation are the normalized bin counts related to different sample spaces. Individually, they can be interpreted as estimated probabilities [of, respectively, voxels of met-PET low and met-PET high uptake coinciding with a given range (bin) of e.g. T1], but their difference do not represent the probability anymore. Neither does the ratio of the difference to the sum. (What would be the corresponding sample space for the left side "probability"? Can probability be negative?)
Following the above arguments, I believe the plot on the right side in the middle panel of Fig. 1 should be corrected. I guess, two upper rows of Fig. 2 are planned to be deleted? The quantities visualised in the plots of the retained lower rows (ABC and DEF) should be rigorously defined in the text.
Fourth, the approach proposed in this paper to finding ranges of T1, T2, and ADC that would indicate areas of high correlation with TDC is, as it was stated in the previous correspondence, "similar" to the methods introduced by Overton [18]. This reference was deleted in the resubmitted version, but it should be included and discussed, to clearly show the difference (in theory and performance) between the two approaches.
Perhaps the work of Chabanas et al. (Analytical Cellular Pathology 13, 1997, pp. 39-47) will also be useful to discuss the main differences. It looks like, in the cited work the cell counts of the histograms were subtracted, bin by bin, instead of the elements of probability histograms. The rationale behind the proposed method should be provided and justified.
Fifth, there are some concerns related to the supplementary material. The sums of "met-PET high" and "met-PET high" probabilities, in columns #2 and #3 of Table 1 are less than 1, by 2% and 4% respectively. They are close to unity for Table 2 and Table 3. Why is that? Moreover, the number of histogram bins in Table 1, 2, and 3 are, respectively, 40, 160, and 60. What is the reason and consequences of these differences? Would they significantly affect the obtained results?
Finally, I suggest considering clarification of the following.
Line 210: What are "histograms of the probability of T1 and T2 relaxation time"? It seems T1 and T2 are independent variables? This is not consistent with "probability histogram of high MET uptake" (line 215).
What is the meaning and measure of "histogram instability" allegedly explaining the uselessness of ADC data (line 244)?
The ranges of T1, T2, and ADC used for correlation analysis in Fig. 3 do not coincide with the yellow-marked ranges in the supplementary Tables 1, 2, and 3. On what basis were the ranges defined actually?
Reviewer 2 Report
Indeed, the authors performed all changes requested.
Author Response
Thank you for your assessment.
This manuscript is a resubmission of an earlier submission. The following is a list of the peer review reports and author responses from that submission.
Round 1
Reviewer 1 Report
The topic of this manuscript is of great importance and the findings seem to be importent as well. My main reservations relate to 1) language and 2) methodology.
1) There are places in this text which are difficult to follow. It is hard to guess the intended meaning of the words/phrases "preferably" (p2, l58), "means" (p2, l83), "integrated diagnosed" (p2, l92-93), "reconstructed" (p7, l238, l240, l242) et cetera. I would suggest asking a native speaker or English language teacher to correct vocabulary and other mistakes.
2) The theory behind the negative probability (Fig. 1, Fig. 2) and histogram subtracting for correlation analysis should be clearly outlined/defined and references to relevant scientific literature should be provided.
Author Response
1) There are places in this text which are difficult to follow. It is hard to guess the intended meaning of the words/phrases "preferably" (p2, l58), "means" (p2, l83), "integrated diagnosed" (p2, l92-93), "reconstructed" (p7, l238, l240, l242) et cetera. I would suggest asking a native speaker or English language teacher to correct vocabulary and other mistakes.
Thank you for your suggestion. We now submit a manuscript that a native English teacher edited.
2) The theory behind the negative probability (Fig. 1, Fig. 2) and histogram subtracting for correlation analysis should be clearly outlined/defined, and references to relevant scientific literature should be provided.
We now provide the following sentence in line 579 to clarify the analytical process performed. We also added a reference that is related to this matter.
“Histogram analysis was also performed using Prism 5 software. A probability histogram was plotted for T1- and T2-relaxation time and ADC. Two histograms were plotted: one for high MET uptake regions (MET-PET high) and another for low MET uptake (MET-PET low). Subsequently, the histogram of MET-PET high was subtracted by the histogram of MET-PET low, which produced a probability histogram that presents the likelihood of the region of interest being MET-PET high as a function of T1- or T2-relaxation time or ADC. Conversely, the negative probability values indicate that the region of interest will be more likely to be MET-PET low. This type of histogram analysis is similar to other scientific research areas, such as data analysis for flow cytometry [18].”
Ref 18; Overton, W.R. Modified Histogram Subtraction Technique for Analysis of Flow Cytometry Data. Cytometry 1988, 9, 619–626, doi:10.1002/cyto.990090617.
Reviewer 2 Report
Dear Authors, I have read with interest the manuscript entitled "Magnetic resonance relaxometry for tumor cell density imaging for glioma: An exploratory study via 11C-methionine PET and its validation via stereotactic tissue sampling".
The work, despite based on a small series, gives very interesting hints on non invasive assessment of tumor cell density, confirmed during multi-region tissue sampling at the time of surgery for glioma. Also, correlation with routinely available imaging sequences highlights the potential of T1 magnetic resonance sequences to reach the informative level of more complex PET studies.
I only have 2 minor suggestions to give:
1) it might be appropriate to better define in the text the criteria followed by surgeons when choosing the sampling sites. It looks intuitive that contrast enhancing areas were avoided but this might be further stressed in the manuscript.
2) I would reccommend revising the acknowledgements section.
Author Response
1) It might be appropriate to better define the criteria followed by surgeons when choosing the sampling sites in the text. It looks intuitive that contrast-enhancing areas were avoided, but this might be further stressed in the manuscript.
Tissue samplings were performed randomly to obtain both contrast-enhancing and non-enhancing lesions. We now provide the number of tissue retrieved from contrast-enhancing or non-enhancing lesions in Table 1 and further added the following sentence in line 556 to clarify the logistics during tissue sampling.
“Tissue sampling was performed to obtain both contrast-enhancing and non-contrast-enhancing tumor tissues. Finally, 25 samples were obtained from contrast-enhanced lesions, and 54 samples were from non-contrast-enhancing lesions.”
2) I would recommend revising the acknowledgments section.
We corrected the acknowledgments section in the revised manuscript.
Round 2
Reviewer 1 Report
Thank you for the text correction. It is clear now.
Nevertheless, the explanations related to my comments on negative probability and the histogram subtraction method are not satisfactory.
Please notice that in the added reference [18] the confusing "negative probability" term is not used at all. As a matter of fact, probability is, by definition, non-negative and not larger than 1. The concept of negative probability, called quasiprobability, is considered in quantum mechanics, but I do not think it is relevant here.
There are four histogram processing methods described in [18] and you claim your analysis method is similar. The question is what exactly was the algorithm you applied. The readers should be provided with the necessary information about that. Otherwise, they would not be able to repeat the described research work. Consequently, the reproducibility principle would not be fulfilled.
The phrase "the histogram of MET-PET high was subtracted by the histogram of MET-PET low" does not make it clear what was the minuend and what was the subtrahend. The confusion is increased by the fact the probability values corresponding to the red and blue histograms in the upper row of Fig. 2 are less that 0.12, but the "subtraction of histogram [red] by histogram [blue]" results in values from the range [-1,0.5]. Moreover, comparing Fig 2. A and B one can see that for T1 in the range from 0 to 800 the bin counts for both MET-PET high and MET-PET low are close to zero, but the mentioned "subtraction" produces the value of -1. One may think that "subtracting by" means something else than "subtracting from". However, one does not know what is the actual meaning of it.
Considering the above, I do not recommend this submission for publication in its present form.